# Fighting Filterbubbles with Adversarial BERT-Training for News-Recommendation

## Abstract

Recommender engines play a role in the emergence and reinforcement of filter bubbles. When these systems learn that a user prefers content from a particular site, the user will be less likely to be exposed to different sources or opinions and, ultimately, is more likely to develop extremist tendencies. We trace the roots of this phenomenon to the way the recommender engine represents news articles. The vectorial features modern systems extract from the plain text of news articles are already highly predictive of the associated news outlet. We propose a new training scheme based on adversarial machine learning to tackle this issue . Our experiments show that the features we can extract this way are significantly less predictive of the news outlet and thus offer the possibility to reduce the risk of manifestation of new filter bubbles. We validate our intuitions in a news recommendation task using a recent attention-based recommendation system.

## 1 Introduction

Recommender systems are a source of unwanted societal effects like filter bubbles or echo chambers, i.e. situations where users reinforce their own beliefs by only engaging with content that corresponds to their world views (Pariser, 2011). This effect is also driven by algorithmic news recommendations, for instance on Facebook (Bakshy et al., 2015). Typically, these automatic recommendations are based on the combination of inferred user preferences and properties of the content: The system learns which properties a user wants to see in the content he consumes. When working with texts, like news articles or social media posts, content properties are often represented by numerical features extracted with large neural network architectures. A prominent example of these neural representations are bidirectional encoder representation from transformers, or BERT (Devlin et al., 2019), which are for instance applied in Google search[1].

We investigate the space of feature vectors induced by BERT models in the domain of news articles and ask if the extraction of features is already a source of societal and cultural divides. We discover that the features are predictive of the news source and hence content recommended based on similarities is likely to come from the same source also, thereby possibly promoting the filter bubble effect. For this study we assume that the news outlet is a good proxy feature for the political orientation or world-view of a given news article. Hence, to address this issue, we propose to apply adversarial learning to compute features that are less predictive of their corresponding news source but are still informative of the semantics of the text. Our adversarial training schemes allow for efficient training of BERT models with adversarial heads.

We begin this paper by briefly discussing the data set and the BERT model we use in this study. Then we describe our novel adversarial training procedure that trains our model, such that the vectorial features it extracts are less predictive of the news outlet. We evaluate our model in a two-fold validation: First we investigate the quality of our new BERT feature representations in isolation using a large corpus of news articles. Second, we investigate the BERT features in a news recommendation setting using a large dataset of real-world user interactions. We conclude our paper with a discussion of our results.

---

[1]https://blog.google/products/search/search-language-understanding-bert/

## 2  RELATED WORK

Adversarial Training has been used in many applications. It was introduces by Goodfellow et al. (**?** in the context of generative image modeling, where a generator neural network generates images that are hard to separate from real images for a discriminator network. Since then, this interplay between two adversaries, the generator and the discriminator, has been used in many appolications. Adversarial learning is a popular choice for tackling many problems with machine learning systems. Prominent examples include learning privacy-aware feature representations, that exclude sensitive private information like gender or race from learned representations. An important subdomain is applying these methods to image data in order to modify images in such a way, that it shadows information like race, age or location from the machine learning systems. Recently, Li et al. 2019 have used an adversarial approach to obfuscate privacy-relevant attributes in images using adversarial learning. But also in other data domains, this becomes increasingly important. The recent work of Lia et al. shows that we can obfuscate privacy-relevant information from user data while maintaining all other relevant information in a feature vector. Prior to this, Pittaluga et al. (2019) have demonstrated a similar result. In our study, this private attribute is the news outlet.

In this study we use the adversarial training to train neural text representations that make it hard to guess their authoring news outlets. Filter Bubbles can also be targeted from the user side. Using techniques from data obfuscation, users can "hide" their true preferences from a system by consuming media from opposing viewpoints. This obfuscation can be done data-driven as demonstrated by and Strucks et al. 2019, where an algorithm suggests the most effective items to interact with to trick the system.

## 3  ADVERSARIAL TRAINING OF BERT MODELS

For feature extraction, we rely on a DistilBERT model Sanh et al. (2019), a more resource-efficient variant of the original BERT deep network architecture. The model takes a sequence of tokens produced by the sentence-piece tokenizer $x_1, x_2, \ldots x_l$ and, in a sequence of 6 transformer layers (Vaswani et al., 2017), produces an output sequence $z_1, z_2, \ldots z_l$ where each $z_i \in \mathbb{R}^{768}$. At the training stage, the model is trained to solve masked language modeling task, where 15% of the input tokens are shadowed and we learn to predict the masked inputs $x_i$ based on the output vector $z_i$.

We use a model by HuggingFace (Wolf et al., 2019) that is pretrained on the Toronto Book Corpus as well as the English Wikipedia . Then we fine-tune this pretrained model on the masked language modeling task using paragraphs from our training data to adapt to the content and style of news articles. This fine-tuned model will act as the baseline in our study.

At inference time – following standard practice (e.g. (Xiao, 2018)) – we compute embeddings of the respective news articles by computing the mean embedding $\bar{z}$ on the output sequence. The BERT model does not process full texts, but only sequences of up to length 512. To mitigate this, we process each paragraph individually, cropping excess tokens, and also average these paragraph vectors into a document vector. This way we reduce the full text, including headline, to a 768-dimensional vector that could be fed into a recommendation system.

We propose a machine learning model comprised of two parts, in accordance with the terminology commonly used in adversarial learning we call them generator and discriminator. The task of the generator model is to produce vectorial features of the tokenized text that contain little information about the news outlet. The discriminator tries to predict the news outlet of the text.

$$f_{\text{gen}}(x_1, \ldots, x_l) = \text{BERT}(x_1, , x_l; \Theta)$$

$$f_{\text{dis}}(z_1, \ldots, z_l) = V^T \sigma(l^{-1} \cdot \sum_{i=1}^{l} W^T z_i)$$

Both models act as adversaries: While the discriminator tries to classify feature vectors correctly by learning $V$ and $W$, the generator learns the BERT parameters $\Theta$ to produce features that allow reconstruction of masked tokens while not allowing the discriminator to correctly classify the output. This is captured in the different optimisation functions associated with the models. The generator

tries to achieve a small loss on the masked-language model task while simultaneously forcing the discriminator to suffer a high classification loss:

$$L_{\text{gen}} = \ell_{\text{mask}}(f_{\text{gen}}) - \ell_{\text{classification}}(f_{\text{dis}} \circ f_{\text{gen}}). \tag{1}$$

On the other hand, the discriminator tries to correctly predict the news outlet by minimising its classification loss:

$$L_{\text{dis}} = \ell_{\text{classification}}(f_{\text{dis}} \circ f_{\text{gen}}). \tag{2}$$

We alternatingly minimize $L_{\text{gen}}$ with respect to $\Theta$ and $L_{\text{dis}}$ with respect to $V, W$. Following Goodfellow et al. Goodfellow et al. (2014), we perform $k$ updates of the discriminator for one update of the generator. Furthermore we pretrain the discriminator for one epoch to give it a head-start and begin the adversarial training with a good discriminator.

There are two variants of training the generator: Instead of using the negative classification loss of the discriminator for the generator training objective (1), we also propose to use the entropy of the classifier

$$L_{\text{ent}} = \ell_{\text{mask}}(f_{\text{gen}}) + \sum_i p_i \log_2 p_i \text{ where } p_i = \text{softmax}\left[(f_{\text{dis}} \circ f_{\text{gen}})(x)\right]_i \tag{3}$$

This method has the advantage of not needing ground truth label information, when the label information is only available on a subset of the data. Then we can train the discriminator on a subset of the data and use all the data for the generator.

## 4 ISOLATED EXPERIMENTS

In our first experimental analysis, we evaluate the adversarial text embedding on its own. We have to investigate two hypothesis: a) The features obtained with adversarial training contain less information about the media outlet than the features from the plain model and b) the features from adversarial training are still useful representations of text.

**Quality Measures** In order to measure these effects we define quantitative measures. To measure how well the features predict the news outlet, we test three different machine learning models on the classification task and report estimates of their performance.
We report the perplexity of

- logistic regression evaluated by 10-fold stratified cross-validation, which resembles the discriminator in our training algorithm,

- a one hidden-layer feed-forward network evaluated by 10-fold stratified cross-validation, which is a more powerful discriminator than used during training, and

- random forest classification also evaluated by 10-fold stratified cross-validation.

The hyperparameters of these evaluation metric classifiers were optimized in a grid search.

To measure of the adversarial features still capture semantic relatedness, we use twoadifferent similarity metrics for texts and check their scores for all nearest neighbor pairs under Cosine similarity of our BERT-feature vectors. We use pretrained Glove word embeddings and compute the Cosine similarity of the resulting document embeddings using the spacy library (Honnibal & Montani, 2017).

**Dataset** Schubert et al. (2018) have crawled a collection of news articles from a wide variety of publishers starting in 2017 up to January of 2020 and kindly have provided us access. We filtered this collection to include only English articles from the 100 most frequent news outlets. Unfortunately we are not at liberty to share this data due to copyright regulations. We split this dataset into training and test data by using all the articles crawled in the last 4 month as well as articles crawled on 33% of the remaining days for testing and the remaining data for training. This way we obtain a training set of 2,998,001 articles and a test set of 377,440 articles.

|  | Only Fine-Tuning | Adversarial Training | Entropy Training |
|---|---|---|---|
| Logistic Regression | 11.23 | **39.71** | 38.76 |
| MLP | 2.62 | **9.00** | 5.07 |
| Random Forest | 15.03 | **37.61** | 26.56 |
| Glove Similarity | **0.9884** | 0.9482 | 0.9559 |

Table 1: Metrics of our BERT features. The adversarial training makes the outlet less predictable by all three classifiers. Simultaneously, the similarity of nearest neighbors judged by Glove-Vectors does not suffer substantially.

**Quantitative Results**    We report our findings in Table 1. We see that the baseline BERT model produces feature vectors that are very predictive of the news outlet: Using a multilayer perceptron the perplexity is 2.62, which indicates a very high chance of correctly guessing the news outlet given the BERT embedding. However, by using the adversarial training, we can dramatically increase the perplexity, making a correct guess much less likely. Interestingly, the quality of the features for judging the semantic similarity does not seem to suffer dramatically. The entropy-based objective is consistently worse than the classification-loss objective. We note that this effect may change upon introduction of a regularization parameter that reweights the two components of the generator losses.

## 5    NEWS RECOMMENDATION EXPERIMENTS

In this section, we analyze the influence of our adversarial news embeddings to the task of news recommendation.

**Model**    To this end we test a state-of-the-art neural network model for news recommendation. The NRMS recommender model (Wu et al., 2020a) is based on attention that has been shown to achieve great performance on the dataset. The model represents each user by a combining its recently clicked news articles using a self-attention layer. Then we model the click probability for a document of a user by the dot product of the user-representation and the document representation. The model is trained with cross-entropy loss where the candidate set corresponds to the classes.

We begin by using a fixed embedding model where the features of the news articles are pre-computed using our BERT models from the last section. We use the adversarial model based on classification loss rather than entropy, as it performed slightly better in the last section. Next, we train the recommendation while simultaneously fine-tuning the embedding model with the adversarial training algorithm. In both cases we represent each news article by its title and abstract. These shorter texts can be embedded on the fly during training.

**Dataset**    We use the Microsoft News Dataset (Wu et al., 2020b), short MIND, for our experiments with news recommendation. The dataset comes in a large and a small variant. The large variant includes 1 million users and 3.38 million behavior logs for training, the small variant includes 50,000 users with 236,344 click behavior logs for training. Each behavior log contains the history of the last clicked news articles of the respective user, as well as the set of candidate news articles with labels that indicate whether the user clicked on the candidate in the given session. For the NRMS training, we process the behavior logs as follows: For each positive example in the candidate set, we sample $K = 16$ negative examples from the log. When there are less than 16 negative examples in the log data, we use the full candidate set. We truncate the history to contain at most 16 news articles.

To obtain the news outlets for the news articles in the dataset, following the original authors' guide, we crawl the articles that are still accessible online using the available crawler tool. We modify the crawler to also extract the news outlet whenever possible. This way we were able to extract news outlet information for a subset of 38983 of the 101,527 news articles in the training data. Unfortunately, presumably due to licensing issues, the datasets features mostly news articles from smaller, non-mainstream media outlets, whereas our study in Section 4 featured mostly larger organizations.

| Metric | Dataset | Fixed Encoder | | End-To-End | |
|---|---|---|---|---|---|
| | | Pre-Trained | Adversarial | Pre-Trained | Adversarial |
| AUC-ROC | small | 0.67 | 0.59 | 0.77 | 0.74 |
| | large | 0.66 | 0.58 | *pending* | *pending* |
| Diversity | small | 4.18 | 4.36 | | |
| | large | 4.82 | 5.23 | *pending* | *pending* |

Table 2: Test AUC-ROC scores and diversity estimates for the different models on the different datasets. We see that the recommendation performance suffers using our adversarial models, however as intended we increase the diversity of the recommendations.

**Quality Measures**    We measure the quality of our recommendations using the area under the ROC curve (AUC-ROC), as proposed by the original authors of the MIND dataset. The adversarial objective and the recommendation objective are at odds when the suggestion of news articles is based on publisher preferences. Hence we can expect a digression of recommendation performance, particularly when the observed data is biased by the currently running system that does not care about reducing the filter bubble effect.

Second, we evaluate the diversity of the news outlets recommended to the users. To this end, for each user in the test data we compute the 100 most probable articles from all of the test articles where a news outlet is known using our recommender model. We measure the diversity of these recommendations by the entropy of the empirical news outlet distributions. A high entropy indicates high diversity among the recommendations, where uniform guessing would yield an entropy of 6.64. We hope to see more diversity in the recommendations of our models trained with the adversarial objective.

**Results**    We summarize our findings in Table 2. As expected we see that the recommendation performance suffers from using our adversarial news embeddings. However, this decrease in performance is rather small. Also as expected, the news recommendation profits from end-to-end training, as this fine-tuning allows the model to learn a representation tailored to the recommendation use-case. Furthermore, we see that the diversity in our recommendations increases with adversarial learning. This illustrates that we have found a new strategy for increasing diversity in news recommendation.

On a side note: Using the BERT model pretrained on the news article corpus improves the performance over using the pre-trained model provided by Huggingface, which achieves a AUC-ROC of 0.76 in the end-to-end setting. This illustrates the importance of appropriate training data for self-supervised representation learning.

## 6    DISCUSSION AND OUTLOOK

We were able to show that features extracted from news articles using modern deep neural networks are highly predictive of the associated news source. With our adversarial training approach we were able to significantly reduce this predictability. However, we are still far away from random guessing. However perfect obfuscation can probably not be achieved without seriously sacrificing the usefulness of the features. For instance, an article on finance is more likely to appear in the Financial Times than other news outlets. In order to hide this fact from the recommendation engine, we have to hide substantial parts of the content, rendering the features pretty much useless.

A caveat of our study are the many hyperparameters in the BERT model, the training process, but also the metrics in our evaluation. We tried to set them according to known best-practices, but right now our results hold only for one particular configuration. Consequently, a more thorough, larger investigation is still required. So far, we have not learned the full recommender engine with the adversarial training routine. This is obviously the logical next step.

In the future we want to also investigate other adversarial objective, for instance learning text representations that are no longer predictive of the general sentiment of a news article. This way we

hope to identify different views on the same news, e.g. from commentators with different political backgrounds.

Finally we should discuss that there are studies that suggest that exposing people to opposing views might in fact further promote division (Bail et al., 2018). This is attributed to the fact that views that are very far away from ones own views are easily dismissed and declared invalid, thereby creating a backfire-effect of further radicalization. To mitigate this, researchers have looked into recommending items that are sufficiently close to the user, but stimulate movement, eg. (Starke et al.). We think this is an interesting research direction for follow up studies.

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
