# OpenReview forum: "Fighting Filterbubbles with Adversarial BERT-Training for News-Recommendation"
_ICLR.cc/2021/Conference — Reject_

### Official Review · AnonReviewer2 · 2020-10-26
**Good motivation, but need to fix many flaws in the paper.**

**Rating:** 3
**Confidence:** 2

**Review:**

This paper proposed an adversarial training framework for reducing the predictive ability of the new outlet from the news recommendation system. While deep learning-based news recommendation systems work very well, these systems tend to recommend contents of the same site, and are likely to develop extremist tendencies.

First, this paper contains many typos and flaws, e.g., the missing references in Section 2, the fluctuation of reference format style, and the different AUC-ROC metric values for the end-to-end setting in the text and table. These flaws make this paper hard to be read. If this paper will be published, I strongly recommend that authors fix these flaws.

Second, while this paper starts with a good motivation, I was not sure that preventing the same outlet from adversarial learning procedures. If we want to recommend news from a large variety of sites, the recommendation service may be able to limit the maximum number of news articles from the same site. Therefore, I was not sure the decrease of the AUC-ROC can be compensated by the diversity increase of sites only. It is better if authors can provide evidence where the proposed adversarial settings can actually prevent filter bubbles, for example, by recommending different opinions than before.

---

### Official Review · AnonReviewer3 · 2020-10-28
**The authors use adversarial training to reduce recommendations of news articles from the same outlet. The effect on filter bubbles is not clear.**

**Rating:** 3
**Confidence:** 4

**Review:**

The authors use adversarial training in an attempt to diversify recommendations of news articles. The effect on filter bubbles is not clear.

The authors address the issue of filter bubbles that according to their rationale result from recommendation engines for news articles. Their assumption is, that a news outlet constantly reports the same opinion and thus users should be recommended articles from a more diverse set of outlets to increase the diversity of opinions consumed by an user. To achieve that, the authors propose adversarial training to generate feature representations that encode outlet-specific information less. They evaluate if their obtained representation is less predictive of the outlet and also how this affects the recommendation performance.

A strong point of this paper is the motivation and application area since it tackles a problem with direct societal impact, something that is typically not attempted in general machine learning papers. The technical novelty is minimal, since only well established techniques are applied. No technical novelty would be acceptable if some insights for the application area are generated. Unfortunately, several naive assumptions are being made and the evaluation goals are insufficient to obtain any valuable results:
1. Outlets are not necessarily biased to one-sided opinions.
2. Forcing recommendations from a variety of outlets can be achieved much simpler. No reasonable baseline is implemented and evaluated.
3. The assumption that content from different sources has higher diversity in embedding space is not necessarily true. Different outlets commonly publish identical texts (for instance copied from the same press agencies).
4. Embedding similarity is not equal to semantic similarity. Bert embeddings capture also syntactic characteristics, which does not necessarily translate to opinions.
6. The data used to measure the recommendation performance is likely biased towards similar outlets. Thus, a performance drop is obvious.

While I really like the general application area and the attempt to tackle such relevant issues, I think the paper is not ready. It needs to be clearer about it's goals, limitations and assumptions. The experimental design needs to be redone to allow to gain any valuable insights. I assume that also a manual analysis is required to confirm any of the hypothesis underlying this research.

---

### Official Review · AnonReviewer1 · 2020-10-29
**This paper studies an interesting problem, fighting filter bubble, by applying GAN in news recommendation. The paper still require some more improvement.**

**Rating:** 4
**Confidence:** 4

**Review:**

The problem, fighting filter bubble, is an interesting and important problem to study for recommender systems. The writing of most parts in this paper are pretty clear and easy to follow. Using adversarial BERT training to solve this problem sounds like an interesting idea.

However, the paper has multiple flaws:
1. Its literature review in Sec. 2 is not complete. It only covers a few work related to adversarial training. In my view, another two lines of work that need to covered are fighting filterbubble/diversity in recommender systems and BERT-based recommender systems.
2. The model is incremental, which is a direct application of adversarial BERT in the news recommendation domain. Its novelty is not enough, especially for ICLR.
3. Experiments of this work are quite limited. No SOTA recommender systems or models to fight filter bubbles are compared. There're no experiments using variants of the proposed model, which left a few questions unanswered. For example, is predicting the news outlet the only task that is effective to train the model? In my view, news outlet only does not represent a user's preference well, so it doesn't really reflect filter bubbles. What will the result be if you use news topic as the target? How about other options? How about using muti-label to train the model?
4. The paper doesn't seem to ready for submission. There're quite a few typos and loose ends. For example, a few citations in Sec. 2  are not in the right format. Several results in Tab. 2 are "pending".

---

### Official Review · AnonReviewer4 · 2020-10-30
**Application of adversarial training for News Recommendation Task**

**Rating:** 5
**Confidence:** 4

**Review:**

Authors propose an adversarial training task for tackling the problem of filter bubbles in news recommendation problem. The main aim of the paper is to propose a method which can obfuscate the news outlet information embedded in the news article's vectorial representation. They propose an adversarial training based method for it.

They propose a standard generator - discriminator setting, where the generator is trying to learn the masked language modelling task and the discriminator is trying to predict the news outlet of the news article. The generator simultaneously tries to optimize for the language modelling task and maximize the news outlet prediction loss. The discriminator tries to minimize the outlet prediction loss.

Strong points of the paper:

1) It tackles a very relevant problem in the current news recommendation problem, i.e. filter bubbles.
2) The paper is well-written with relevant experiments.

Weak points:

1) The paper is a simple application of the well-established adversarial training methods and there is very little technical novelty in terms of the core contributions. Would advise authors to submit this work in an applied data sciences conferences like SIGIR, CIKM, ECIR etc.

2) Authors have clearly put in a lot of efforts and they are praiseworthy, but given the nature of the contributions, submission to a more applied conference is suggested.

---

### Decision · Program_Chairs · 2021-01-07
**Final Decision**

**Decision:**

Reject

**Comment:**

All reviewers agree that this paper is not ready for publication. In addition to the technical comments, the authors should pay attention to the comments by Reviewer 3 about the naivete of the motivation provided for the work. Filter bubbles (to the extent that they really exist; there is controversy about this) have multifactorial origins.